# Gold and Iron Oxide Nanoparticle Assemblies on Turnip Yellow Mosaic Virus for In-Solution Photothermal Experiments

**DOI:** 10.3390/nano13182509

**Published:** 2023-09-07

**Authors:** Ha Anh Nguyen, Sendos Darwish, Hong Nam Pham, Souad Ammar, Nguyet-Thanh Ha-Duong

**Affiliations:** 1Phenikaa University Nano Institute (PHENA), Phenikaa University, Yen Nghia, Ha Dong, Hanoi 12116, Vietnam; anh.nguyenha@phenikaa-uni.edu.vn; 2Laboratoire ITODYS, CNRS UMR-7086, Université Paris Cité, 15 rue J-A de Baïf, 75013 Paris, France; sendos.darwish1@gmail.com (S.D.); souad.ammar-merah@u-paris.fr (S.A.); 3Institute of Materials Science, Vietnam Academy of Science and Technology, 18 Hoang Quoc Viet, Cau Giay District, Hanoi 10000, Vietnam; namph@ims.vast.ac.vn

**Keywords:** hyperthermia, plant virus, plasmonic nanoparticle, magnetic nanoparticle

## Abstract

The ability to construct three-dimensional architectures via nanoscale engineering is important for emerging applications in sensors, catalysis, controlled drug delivery, microelectronics, and medical diagnostics nanotechnologies. Because of their well-defined and highly organized symmetric structures, viral plant capsids provide a 3D scaffold for the precise placement of functional inorganic particles yielding advanced hierarchical hybrid nanomaterials. In this study, we used turnip yellow mosaic virus (TYMV), grafting gold nanoparticles (AuNP) or iron oxide nanoparticles (IONP) onto its outer surface. It is the first time that such an assembly was obtained with IONP. After purification, the resulting nano-biohybrids were characterized by different technics (dynamic light scattering, transmission electron microcopy, X-ray photoelectron spectroscopy…), showing the robustness of the architectures and their colloidal stability in water. In-solution photothermal experiments were then successfully conducted on TYMV-AuNP and TYMV-IONP, the related nano-biohybrids, evidencing a net enhancement of the heating capability of these systems compared to their free NP counterparts. These results suggest that these virus-based materials could be used as photothermal therapeutic agents.

## 1. Introduction

Hyperthermia therapy is one of the most promising strategies to treat cancers, either as an alternative or as a complement to chemotherapy and radiotherapy. Hyperthermia is defined as moderate elevation in temperature in the range of 42 to 45 °C, leading to the destruction of malignant cells with minimal injury to adjacent healthy cells [1,2]. The elevation in temperature affects the viscosity and stability of cell membrane, which may alter transport/receptors functions and cells’ signaling mechanisms. It also inhibits DNA repair functions. However, proteins seem to be the primary target of hyperthermia. Indeed, it may modify cellular homeostasis, post-translational modifications, induce aggregations of denatured proteins, and increase oxidative stress [2]. Heat may be induced by several external physical sources, such as microwaves, radiofrequency, ultrasounds, magnetic fields, and lasers [1].

Cell-uptaken plasmonic nanoparticles, such as gold nanoparticles (AuNPs), excited by a laser light induce local hyperthermia, known as photothermia [3,4]. As excellent light absorbers, due to their plasmon resonance and heat conductors [5], they can be accumulated preferentially in tumors by the enhanced permeability and retention (EPR) effect or by tumor-specific delivery. Similar features may be observed with ceramic nanoparticles, such as iron oxide nanoparticles (IONPs), replacing plasmon resonance as origin of heating by phonon excitation of the ceramic crystallographic lattices [6]. In both cases, the biocompatibility, the chemical stability, and the dispersibility in biological media make such laser light-induced hyperthermia particularly suitable. The use of appropriate laser power allows efficient light to heat energy conversion while minimizing healthy tissue damage and side effects [7]. The selection of laser wavelengths within the biological spectral windows (700 to 1400 nm) optimizes optical penetration depth into tissues [4]. AuNPs can interact with the light of a specific wavelength and convert it into thermal energy in a short time; consequently, the local temperature increases [8]. Their plasmon absorption wavelength can be tuned in the near-infrared (NIR) (650–940 nm) and infrared (1000–1350 nm) region by simply acting on their morphology (size and shape) and assembly [9]. Gold nanocages [10,11], rod-shaped AuNPs [12,13,14], silica-gold core-shell [15], raspberry-like gold nanostructures [16], etc. exhibit high photothermal heat conversion under near-infrared laser irradiation. By clustering or aggregating AuNP, photothermal activity is often significantly enhanced [17,18,19,20,21]. IONPs also show light-to-heat conversion, depending on their light-induced lattice vibration in the same spectral range [22,23,24,25]. However, the photothermal response rate and the final temperature could be low compared to AuNP. Also, as nanoclusters and/or nanoaggregates, their molar absorption coefficient in the near-infrared region can then be strongly improved, making them also valuable for photothermal therapy [22].

Since the aggregation state of these two types of particles appears as an issue for photothermia, and to improve the photothermal properties of individual nanoparticles, we used in this work plant viruses as 3D scaffolds to control the assembly of AuNPs and IONPs. In the last decades, plant virus-based nanomaterials have been developed for many biomedical applications, such as drug and gene delivery, vaccine development, theragnostics, etc. Thanks to their remarkable properties, such as stability, highly defined particle morphology, ease of production, capsid surface modification, and non-toxicity for mammals [26,27,28,29,30], they are nowadays deeply investigated. For example, in cancer treatment, Johnson grass chlorotic stripe mosaic virus (JgCSMV) [31], tobacco mosaic virus (TMV) [32], and potato virus X (PVX) [33] are exploited as nanocarriers for doxorubicin, cisplatin, and methyl auristatin, respectively. 

To the best of our knowledge, there are no examples of AuNPs and IONPs assembled around plant viruses; neither are there examples of the use of the resulting assemblies for hyperthermia application. As a pioneer study, the grafting of IONPs and AuNPs onto turnip yellow mosaic virus (TYMV) and the evaluation in solution of their photothermal properties is addressed here. TYMV is an icosahedral plant virus capsid of about 28 nm in diameter. Its capsid is composed of 180 self-assembled chemically identical protein subunits arranged in a T = 3 icosahedron [34]. TYMV capsid is stable at high temperatures, in a large range of pH, and in the presence of organic solvents, allowing its functionalization by several reactions such as EDC/NHS amidation or click chemistry reactions [35,36]. It is completely safe for humans and has already successfully applied as a template for nanoparticle assembly [37]. In this article, we show that the resulting TYMV-based nano-biohybrids are soluble in biological media and possess enhanced photothermal effects compared to free NPs.

## 2. Materials and Methods

### 2.1. Plant Virus Extraction and Purification

TYMV was extracted and purified as previously described [37,38].

### 2.2. Synthesis of Citrate-Coated IONPs

IONPs of about 10 nm in size were synthesized by forced hydrolysis in polyol [39]. In practice, 4.34 g (0.01 mol) of iron(II) acetate was mixed with 0.11 mol of distilled water in 250 mL of triethyleneglycol in a tri-neck balloon. The mixture was sonicated for 10 min to homogenize the red dispersion, then stirred at 500 rpm, heated up to 220 °C with a heating rate 6 °C min^−1^, and maintained under reflux for 2 h. After cooling down to room temperature, the precipitated dark brown powder was collected by centrifugation at 15,000 rpm/20 min. The solution was then washed twice with acetone, to remove the excess of polyol, and twice with hot water, to oxidize the black magnetite Fe_3_O_4_-type particles into brown maghemite γ-Fe_2_O_3_ ones [40]. Finally, the recovered powder was dried in air at 50 °C overnight.

IONPs were then functionalized with citric acid. In a two-necked flask, 500 mg of the as-produced IONPs were dispersed in 25 mL of water by sonication, and 1 g of citric acid (CA) was added to the dispersion. The reaction took place under continuous stirring at boiling temperature for 30 min to ensure the complete displacement of the adsorbed polyol from the particle surface by CA. Once completed, a magnetic decantation followed by immersion in ethanol was used to remove the excess ligand. Finally, functionalized particles were dried in ambient air overnight and dispersed in deionized water on demand.

### 2.3. Synthesis of Citrate-Coated AuNPs and Functionalization

AuNPs of about 10 nm in size were synthetized according to the procedure described by Turkevich et al. [37,41,42]. Briefly, to 150 mL of boiling water, a solution of chloroauric acid (HAuCl4) at 1.11 mM and 15 mL of trisodium citrate at 69.36 mM were added. The mixture was stirred whilst boiling for 15 min. AuNPs were then functionalized by adding 1-amino-6-hexanethiol (AHT) for the grafting onto TYMV (500 coating molecules for 1 AuNP).

### 2.4. NPs Grafting onto TYMV

CA- and AHT-, respectively, coated IONPs and AuNPs were grafted onto TYMV through an EDC/NHS coupling reaction. For IONPs, 1-ethyl-3-(3-dimethylaminopropropyl)carbodiimide hydrochloride (EDC, 2 mM) was added to activate the carboxyl group of their surface citrate groups (5 mg in 250 µL). The solution was incubated at room temperature (RT) for 20 min, followed by the addition of N-hydroxy-succinimide (NHS, 5 mM) in water. After 40 min, the solution was washed to eliminate the excess of EDC and NHS and the pre-activated carboxylic IONPs were then mixed with TYMV (10.4 µL at 0.1 mg/mL) during 48 h at 20 °C (800 rpm, thermomixer, Eppendorf, France). The grafting of AuNPs onto viruses was as already described for TYMV [37]. We used, here, the first strategy described. Briefly, EDC (2 mM) was added to activate the carboxyl group on the surface of the TYMV capsid (100 µg/mL) and incubated at RT for 15 min. This was followed by the addition of NHS (5 mM). After 30 min, the pre-activated carboxylic TYMVs were reacted with AuNPs functionalized with AHT at RT in the dark, overnight. Whatever the grafting strategy, the separation and purification of the resulting nano-biohybrids were performed by agarose gel and agarose digestion, as described in our previous work [37].

### 2.5. Characterization of Nano-Biohybrid Materials

In order to confirm the grafting, the newly synthetized nano-biohybrid was characterized by X-ray photoelectron spectroscopy (XPS), transmission electron microscopy (TEM), and dynamic light scattering (DLS), UV-visible absorption spectrometry. XPS was performed on a KAlpha+ system (Thermo Fisher Scientific, East-Grinstead, UK) equipped with a micro-focused and monochromatic Al Kα X-ray source (1486.6 eV, 12 kV) with an X-ray spot size of 400 μm. TEM was carried out after depositing drops of the nano-biohybrid’s aqueous solutions onto carbon-coated copper grids using a JEM 2100F plus microscope (JEOL, Tokyo, Japan), operating at 200 kV [43]. ζ potential and hydrodynamic diameter measurements were performed on a Malvern Nano Zetasizer setup (Malvern Panalytical, Worcestershine, UK), operating with a laser of 633 nm, on colloidal dispersions of the nano-biohybrids in distilled water.

### 2.6. Photothermal Experiments

Prior to photothermal experiments, aqueous nano-biohybrid solutions were prepared and their iron or gold mass concentration was determined by X-ray fluorescence spectroscopy (XRF). A Minipal4 spectrometer (Panalytical, Almelo, The Netherlands) equipped with a Rh-Kα tube (30 kV, 87 μA) was used and the quantification was achieved via pre-plotted calibration curves using Fe^3+^ standard solutions. The gold mass concentration was also calculated according to the protocol described in [37]. Photothermal experiments were conducted using a laser irradiation (LD808E3WG13, Lasermate group, Inc., Walnut, CA, USA) at a wavelength of 808 nm with a varying power density (from 0.165 to 1.5 W/cm^2^). For each experiment, a volume of 200 µL of the freshly prepared aqueous colloids of free IONPs and AuNPs, on one hand, and of their TYMV-based hybrids, on the other hand, was considered [44].

## 3. Results and Discussions

### 3.1. Characterization of IONPs and AuNPs

The hydrodynamic size of naked IONPs in water measured by DLS shows a hydrodynamic size of 200 nm due to magnetic attraction, causing aggregation of the nanoparticles. Moreover, the zeta potential of IONPs exhibits a ζ value of −3.6 mV at pH 6.5, suggesting low stability of the colloid and explaining their tendency to aggregate. Thanks to their citrate functionalization, their hydrodynamic size and their ζ value decreased down to 10 ± 5 nm and −60 ± 5 mV, respectively (Figure 1A and Appendix A), in agreement with high colloidal stability, which is mandatory for the target goal of this work. The prepared aqueous CA-coated IONP-based suspension was stable, even after one month without any aggregation evidence. The particle size was found to be coherent with that deduced from TEM observations. The collected micrographs on the citrate-coated particles evidenced spherical-in-shape single crystals (Figure 1B). The particles exhibit an almost narrow size distribution with an average diameter of 12 nm for a standard deviation of 2 nm (normal law statistics). XPS analysis confirmed that the produced particles are consistent with the maghemite phase. Fe 2p spectra collected on both bare IONPs and citrate coated ones exhibit typical Fe 2p_3/2_, Fe 2p_1/2_, and Fe 2p_3/2_ satellite signatures at 710, 724, and 718 eV, respectively (Figure 1C). The difference in binding energy between the Fe 2p_3/2_ peak and the satellite peak is approximately 8 eV, which is associated with Fe^3+^, as reported in the literature on maghemite [45]. Additionally, the presence of citrate on the surface of the functionalized particles was confirmed by analyzing the C1s signal. For both systems, C 1s spectra were deconvoluted within typical C-H/C-C, C-OH, and O=C-OH signatures at 285.0, 286.4, and 288.5 eV, respectively [46] (Figure 1D), and the total C/Fe atomic ratio increased for the surface modified particles compared to the bare ones (from 0.72 to 1.12). XRD and FTIR experiments confirmed also the functionalization of IONP by CA (see Appendix A).

The same investigations were performed on AuNPs@CA and AuNPs@ATH [37]. DLS and TEM measurements confirmed that we succeeded to synthesize, by the Turkevich route, spherical particles of about 10 nm in size (Figure 2A,B). These particles are mainly metallic gold particles covered by citrate species, as confirmed by XPS (Figure 2C–E), which can be easily replaced by AHT, as the affinity of gold to thiols is higher than that of carboxylates. Typically, for both kinds of particles, the XPS Au 4f signal is consistent with a 4f_5/2_ and 4f_7/2_ doublet of Au^0^ at 87.7 and 84.0 eV, respectively [47]. XPS N 1s spectra showed an increased signal for the particles grafted to AHT: a broadened peak at 400 eV, characteristic of amino/ammonium groups, was evidenced [48]. Similarly, only the S 2p XPS spectrum of coated particles exhibits a typical signature of sulfur on the AuNPs@ATH surface [49,50]. The peaks at 161.9/163.1 eV, 164.0/165.2 eV, and 167.9/169.1 eV can be easily assigned to the S 2p_3/2_ and S 2p_1/2_ signals of thiol in interaction with gold (grafted), free thiol, and oxidized thiol, respectively, with the major one being the first.

### 3.2. Characterization of the Nano-Biohybrids

After activating the –COOH of TYMV, AuNPs@AHT-bearing amino groups were attached by EDC/NHS coupling, and, reversely, the carboxylate groups of IONP@CA were activated to react with the amino groups of TYMV. Indeed, each subunit of TYMV capsid contains 4 acidic amino acids (Glu or Asp) and two lysines accessible at the exterior surface [51], enabling the attachment of inorganic particles by one route or another. In both cases, after grafting, the unreacted IONPs@CA, AuNPs@AHT, and TYMV were separated from IONPs and AuNPs covalently bound to TYMV (TYMV-IONP and TYMV-AuNP) using agarose electrophoresis (Figure 3A,B, respectively). The gel was stained with Coomassie blue to detect the TYMV capsid. For both systems, the iron oxide or gold grafted onto TYMV migrated slower than free TYMV and single nanoparticles. In this way, the bands corresponding to the produced nano-biohybrid were digested by agarose, as previously described [37].

As expected, DLS experiments indicate an increase in diameter for the newly synthetized hybrids compared to free TYMV and IONPs@CA alone (Figure 4A), on one hand, and compared to free TYMV and AuNPs@AHT alone (Figure 4C), on the other hand. The hydrodynamic diameter was found to be 80 ± 30 nm for TYMV-IONPs, which is higher than that found with TYMV-AuNPs [37]. TEM micrographs recorded on the separated TYMV-IONPs and TYMV-AuNPs confirmed the almost complete absence of aggregates, other than those templated by TYMV capsids, as well as the almost complete absence of free IONPs and AuNPs. This indicates that our surface engineering strategy is an elegant way to assemble in 3D, whether gold or iron oxide particles, within controlled arrangements. TEM also confirmed the presence of IONPs and AuNPs around the icosahedral TYMV, highlighting the particle arrangement along a five-fold symmetry (Figure 4B,D).

For simplification, IONPs@CA and AuNPs@AHT colloids are labelled in the further sections as IONP and AuNP colloids.

### 3.3. Evaluation of Photothermal Properties

To check the photothermal capabilities of the engineered TYMV-IONP and TYMV-AuNP assemblies in water, the focus was made on the reached equilibrium temperature (after 5 min of laser irradiation), the specific absorption rate (SAR), which measures the power dissipated per gram of metal element, and the photothermal conversion efficiency η.

The SAR values were experimentally determined using the Equation (1) [23]:(1)$$\mathrm{SAR}=\frac{C\times m_{s}}{m_{Me}\;}\times \frac{dT}{dt}$$
where *C* is the specific heat capacity of the studied samples, namely TYMV-IONP and TYMV-AuNP aqueous colloids, and is assumed to be equal to that of water (*C* = 4.185 J·g^−1^·K^−1^), *m_Me_* is the total mass of iron or gold in the sample, *m_s_* is the total mass of the sample, and *dT*/*dt* is the slope in the first 30 s of the temperature increase vs. time.

The photothermal conversion efficiency *η* was calculated according to Equation (2) [52]:(2)$$\eta =\frac{B\left(T_{ss}-T_{0}\right)\rho_{w}c_{w}V_{w}}{I\left(1-10^{-A_{\lambda }}\right)}$$
where *I* is the incident laser power (0.41 W/cm^2^), *A_λ_* is the absorbance of AuNP solution at 808 nm, *V_w_* is the volume of the solution (200 µL), *ρ_w_* is the water mass density (1 g/mL), *c_w_* is the specific heat capacity of water (4.185 J·g^−1^·K^−1^), *T_ss_* is the temperature at steady state, and *T*_0_ is the initial temperature. *B* (s^−1^) is the rate constant of heat loss, which is determined by plotting [52]:(3)$$ln\frac{T-T_{0}}{T_{max}-T_{0}}=-Bt\;\mathrm{with}\;t\;\mathrm{is}\;\mathrm{time}\;(s)$$

This rate constant *B* of heat loss is estimated to be 0.007 s^−1^ for AuNP-TYMV.

The optical absorption of the TYMV-IONP and TYMV-AuNP aqueous colloids at the laser excitation wavelength of the photothermia setup was determined. We started by recording the absorption spectra of the TYMV-IONP, TYMV-AuNP, IONP, and AuNP aqueous colloids (Figure 5A,B). For comparison, adapting the previous gold synthesis protocol, we were able to prepare 20 nm-sized gold particles, as well as their related TYMV hybrids [37], and we built their assembly around the TYMV virus, using the same experimental protocol as described previously. The optical absorption spectra were thus collected on their aqueous colloid before photothermal assays.

For the iron oxide-based colloids, an increase in absorption was observed for the particle assembly (Figure 5A). Consequently, an improved photothermal effect was expected on the clustered particles, as already reported [53]. In the present case, after only 5 min of irradiation at 808 nm (0.41 W/cm^2^), the temperature of the solution containing free IONPs (0.125 mg/mL of Fe) increased by 3 °C, with the equilibrium temperature of 3.5 °C reached after 10 min of irradiation. The higher the iron concentration, the higher the temperature [54]; the temperature reached more than 15 °C for a concentration for 1 mg/mL (Appendix A). When IONPs were assembled around the virus capsid, the change in temperature increased to 5.6 °C (0.125 mg/mL of Fe), with the equilibrium temperature of 6.9 being reached after 10 min of irradiation (Figure 6A). When the laser is off, the temperature decreases and returns to its initial value. Similar to free IONPs, as the iron concentration increases from 0.125 mg/mL to 1.00 mg/mL, TYMV-IONP samples exhibit a more important temperature increase: from 5.6 to 13.8 °C after 5 min, reaching the equilibrium after more than 10 min at slightly higher temperature values (Figure 6B). Finally, the temperature changes vary with the laser power density (Figure 6C, Table 1). Because living cells are able to self-repair the damages induced by heat, the laser irradiation was repeated several times [55]. In practice, the studied colloids were irradiated at 808 nm during 5 min (laser on). When the temperature decreased to room temperature after the laser was turned off, a new irradiation cycle was applied, and this process was repeated for a total of two “on-off” cycles, measuring, at each cycle, the photothermal response (Figure 6D). The recorded data clearly confirmed that TYMV capsid can resist a temperature rise of at least 16.9 °C.

For the IONP colloid, these temperature increases are comparable to those reported in the literature for almost similarly sized, citrate-coated maghemite nanoparticles in water [54,56], while those measured on the TYMV−IONP colloid (Table 1) are still higher than those reported on 25 nm iron oxide nanoflowers for equivalent iron concentrations, in water, too [56]. Indeed, the temperature change induced by IONP’s assembly around TYMV is similar to the one observed for iron oxide nanoflowers of about 25 nm in size, but with an iron concentration 10 times lower, or to IONPs attached to human serum albumin (HSA) with a lower power density. This suggests that our engineered hybrids may operate as heating agents at low iron concentration, limiting transient perturbations of iron metabolism [57], and, at low laser power density, avoiding any laser side effects on healthy tissues due to overexposure.

SAR values of free and TYMV-assembled 10 nm-sized iron oxide particles were also measured and compared to those reported in the literature (Table 1). Surprisingly, despite a significant temperature increase, the obtained SAR values on IONP aqueous colloid were found to be smaller than those reported in certain studies [54], even at high laser power density, but comparable to those of others [56,58]. In all the cases, the values measured on TYMV−IONP colloid were found to be higher than those determined on free particles, in good agreement with Chen et al.’s conclusions on the beneficial effect of particle clustering [59]. Note that the discrepancies between our SAR values and those reported in certain literatures may also be due to the phase difference between our particles and those described in the related papers. Indeed, some authors use iron oxide terminology without defining the maghemite or magnetite nature of their particles. Others specifically refer to maghemite particles, while others focus on magnetite ones. These differences are not trivial at all. Indeed, despite their structural and chemical proximity, the unit cell lattice of maghemite is smaller than that of magnetite (8.36 vs. 8.40 Å), with smaller iron cation–oxygen anion distances, meaning lattice stretching vibrations at higher wavenumbers. Also, the former contains only ferric cations, while the latter contains both ferrous and ferric cations, making it darker (black vs. red brown), able to significantly absorb the visible light and to increase the photothermal properties [23]. nanomaterials-13-02509-t001_Table 1Table 1Photothermal characteristics of free and assembled on TYMV iron oxide particles in water compared to those of the literature.Particles[Fe](mg/mL)Laser Power Density (W/cm^2^)ΔT (°C) along 10 min of ExpositionSAR (W/g_Fe_)Ref.Spherical free 10 nm-sized maghemite particles0.1250.413.5568*1.000.4115293*10 nm-sized maghemite particles assembled on TYMV0.1250.416.7742*1.000.27.4184*1.000.4116.2585*1.000.618.9523*10 nm-sized maghemite grafted to HSA0.141.09.21870[60]Iron oxide nanoflowers, 25 nm in size1.40.32.5~180[56]1.41.08~550* This study.


For the gold-based colloids, a red-shift in the plasmon resonance band was evidenced in the spectrum of the particle assemblies, indicating a higher absorption coefficient at 808 nm for the assemblies compared to the free particles (Figure 5B). Consequently, free AuNPs would exhibit smaller photothermal effects compared to assembled ones on TYMV. This is exactly what we observed (Figure 7A). Indeed, when fixing the laser power density to 0.41 W·cm^−2^, no increase in temperature was observed in pure water after 5 min of laser exposure. However, a temperature increase of 5 and 8 °C after 5 min of exposure was observed in free and TYMV-assembled AuNPs, respectively. Interestingly, the reached temperature changes on the resulting aqueous colloids were found to be more important, when 10 nm-sized AuNPs assembled on TYMV are replaced by 20 nm-sized ones. The temperature increased by 4 and 12 °C after 5 min of laser irradiation for free and assembled particles, respectively. But, in all the cases, when the laser is turned off, the temperature drops rapidly to room temperature (Figure 7A).

Focusing on these nano-biohybrids, when the laser power density was increased from 0.165 to 1.5 W·cm^−2^, the change in temperature rose from 7.7 to 48.7 °C, exhibiting a linearity between the photothermal effect against the power density (Figure 7B).

A total of six “on-off” cycles of laser exposure on TYMV−AuNP hybrids, regardless of the size of the involved gold particles, 10 or 20 nm, were also applied, demonstrating the robust stability of the photothermal properties (Figure 7C,D). This indicates the virus capsid’s stability, as they were not denatured or disassembled during the heat processing. This result is not surprising, since plant virus capsids are known to be stable, even at high temperature [28].

The calculated photothermal conversion efficiency is 66% and 88% for TYMV-AuNP 20 nm and TYMV-AuNP 10 nm, respectively, in agreement with previous reports that smaller AuNPs have larger photothermal conversion efficiency [61,62]. In addition, the SAR values were determined from the temperature curve for the four systems (Table 2). Compared to the literature, the η values measured on TYMV-AuNP solution were found to be of the same order as those already reported for larger free particles [52,61] or similarly sized particles, but irradiated with a higher laser power density [59]. Similar trends were observed when comparing SAR values to those reported in the related literature. It was demonstrated that, SAR values increase when the gold concentration decreases, whereas, the lower the gold concentration, the lower the equilibrium temperature [63,64]. In our experiments, the gold concentrations were very low (around 20 ppm), explaining the high SAR values of free AuNP colloidal dispersion in water, even though the temperature changes were weak. However, for assembled AuNP colloids, these values reached 22.7 and 57.4 kW/g_Au_, under the same conditions, which corresponds to an increase factor of 2.5 to 4.5 for 10 nm-AuNP TYMV and 20 nm-AuNP-TYMV, respectively.
nanomaterials-13-02509-t002_Table 2Table 2Photothermal characteristics of free and assembled on TYMV gold particles in water compared to those of the literature.ParticlesLaser Power Density (W/cm^2^)ΔT (°C) along 5 min of Exposition[Au]mg/mLSAR (kW/g_Au_)η (%)Ref.Spherical free 10 nm-sized Au particles0.4150.0198.2 ± 0.4Nd*10 nm-sized Au particles assembled on TYMV0.4180.01922.7 ± 0.988*Spherical free 20 nm-sized Au particles0.4140.02412.6 ± 1.7Nd*20 nm-sized Au particles assembled on TYMV0.41120.02457.4 ± 3.466*Nanoraspberry of 41 nm0.30150.1566[16]Nanorod 40 × 15 nm0.3120.1465Nanorod 55 × 15 nm0.3200.18.565Nanocluster1.2400.02nd83.7[59]* This study, (nd) not determined.

Comparing now TYMV-IONP and TYMV-AuNP biohybrids, the assembly of AuNPs of similar size exhibited a 30-fold higher SAR value (Figure 8) and reached the equilibrium temperature more rapidly (5 vs. 10 min). Our results are consistent with those of several other groups, which have demonstrated that gold presents better photothermal properties than iron oxide nanoparticles, and that, at low concentrations of metal, plasmonic nanoparticles can deliver more efficient therapeutic heating [56,58]. This is particularly true when the light absorption capability of gold particles is adjusted around the operating photothermal light excitation wavelength. The light absorption properties of iron oxide particles offer less tunability.

## 4. Conclusions

In this work, we succeeded to synthesize and characterize IONP and AuNP assembly around virus capsids. We showed that the use of TYMV capsid is an elegant way to organize, in 3D, functional nanoparticles of different chemical natures and properties (metallic gold and ceramic iron oxide), and to obtain new soluble and stable materials in biological media. We then performed photothermal experiments in solution with these assemblies in comparison to free particles. We showed that the capsid-templated 3D-organization of these particles enhanced their intrinsic properties. The resulting nanohybrids exhibited both higher equilibrium temperature and SAR values, making their therapeutic use possible with lower metal concentration, especially with AuNP assemblies. This study serves as a proof of concept on the use of TYMV capsid to assemble inorganic nanoparticles and to improve their photothermal response. TYMV can be replaced by any other plant virus and Au and Fe_2_O_3_ NPs can be exchanged by any other inorganic particles to develop new nano-biohybrids for nanomedicine application. Obviously, in cellulo and in vivo assays must be performed in a second and third step before these hybrids can be considered as photothermal therapeutic agents in clinical settings.

## Figures and Tables

**Figure 1 nanomaterials-13-02509-f001:**
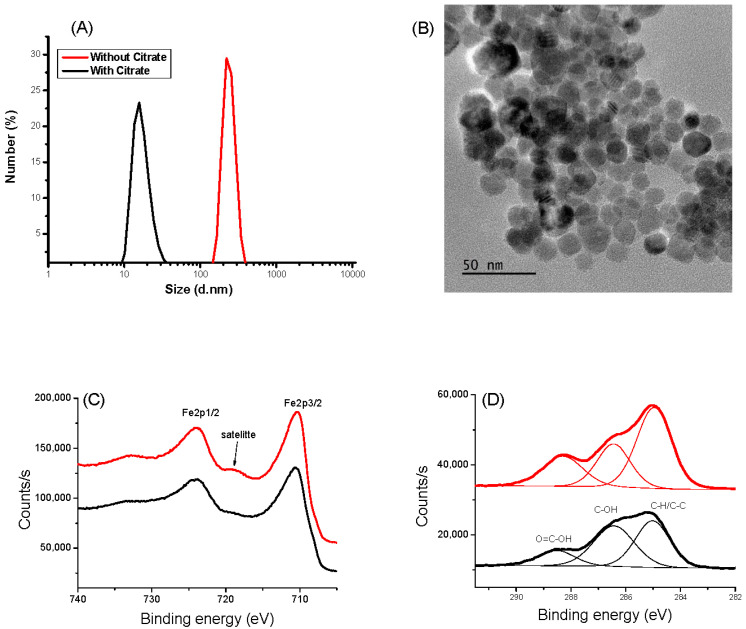
Characterizations of IONP (black) and IONP@CA (red). (**A**) DLS measurement. (**B**) TEM images: XPS spectra of IONP and IONP@CA. (**C**) HR-Fe 2p region. (**D**) HR-C 1s region.

**Figure 2 nanomaterials-13-02509-f002:**
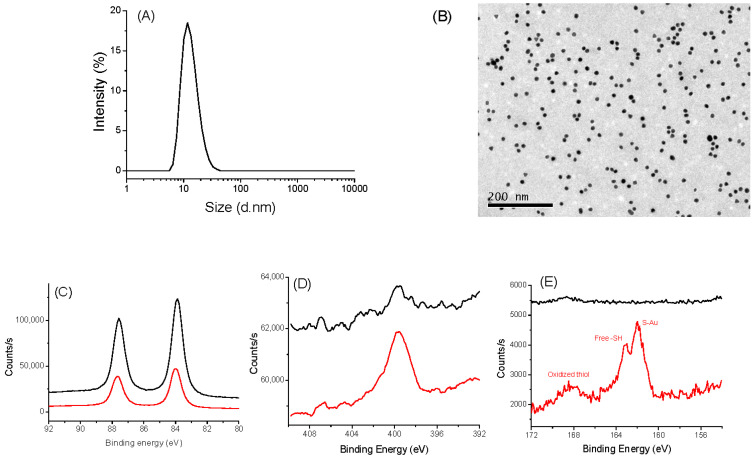
Characterizations of AuNPs@CA (black) and AuNPs@AHT (red). (**A**) DLS measurement of AuNPs@CA. (**B**) TEM images AuNPs@CA: XPS spectra of AuNPs@CA (black) and AuNP@AHT (red). (**C**) HR-Au 4f region. (**D**) HR-N 1s region. (**E**) HR-S 2p region.

**Figure 3 nanomaterials-13-02509-f003:**
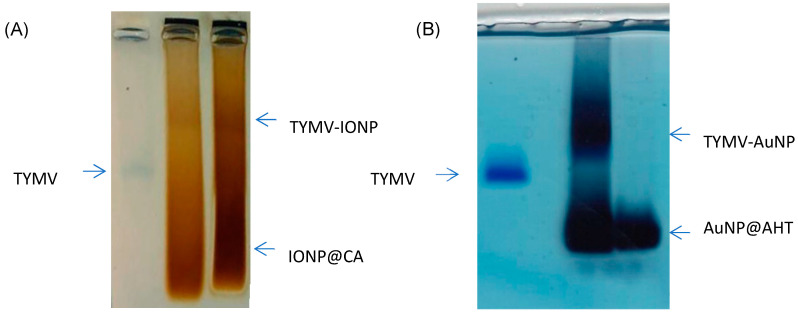
Agarose electrophoresis of (**A**) free TYMV, IONPs@CA, and TYMV-IONP, and of (**B**) free TYMV, TYMV-AuNP, and AuNPs@AHT for optimum grafting and separation conditions.

**Figure 4 nanomaterials-13-02509-f004:**
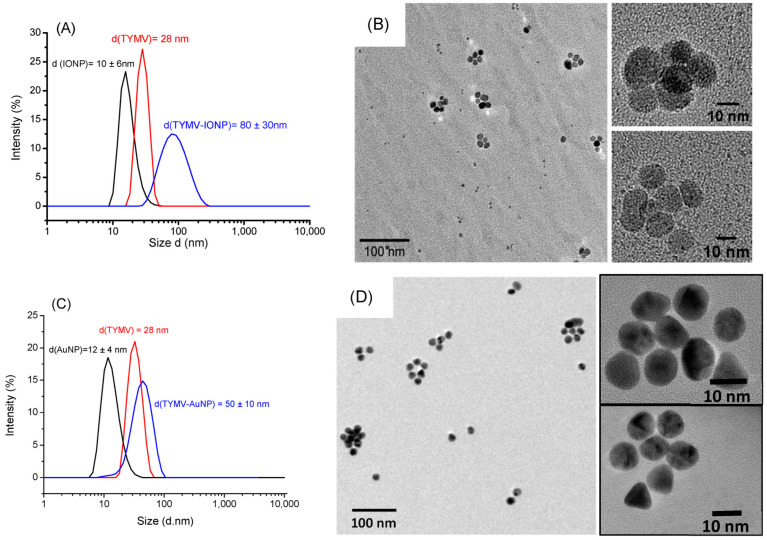
DLS measurement (**A**,**C**) and TEM pictures (**B**,**D**) of TYMV-IONP and TYMV-AuNP, respectively, after agarose digestion, highlighting the formation of icosahedral particle arrangements.

**Figure 5 nanomaterials-13-02509-f005:**
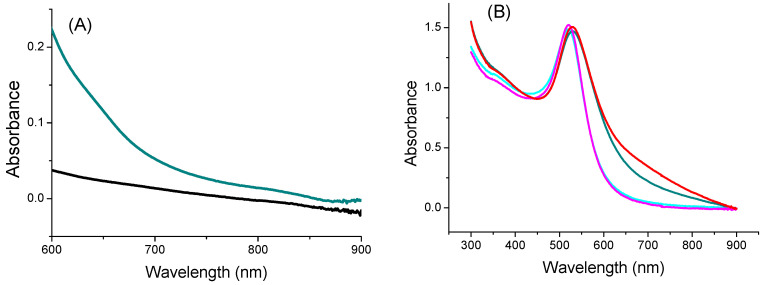
UV-Visible-NIR absorption spectra of (**A**) IONP (black) and TYMV-IONP (dark cyan) and of (**B**) AuNP (cyan) and TYMV-AuNP (dark cyan). For comparison, the spectra of 20 nm-sized AuNP@AHT colloids (magenta) and their related nano-biohybrids (red) are also given.

**Figure 6 nanomaterials-13-02509-f006:**
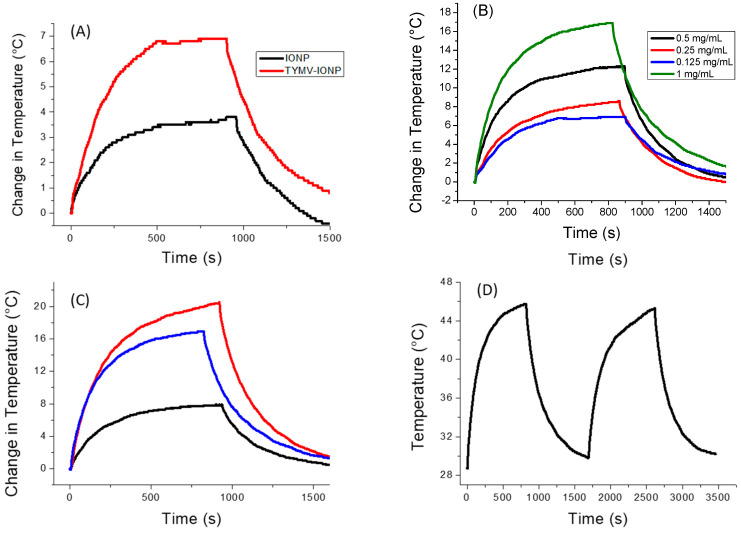
Photothermal experiments on IONP and TYMV−IONP aqueous colloids. (**A**) Temperature curve of IONP and TYMV−IONP colloids (0.125 mg_Fe_/mL) under the excitation of 808 nm with laser on–laser off (0.41 W/cm^2^). (**B**) Temperature curve of TYMV−IONP colloid at different iron concentrations. (**C**) Temperature curve of TYMV−IONP colloid (1 mg_Fe_/mL) under different power density laser at 808 nm: 0.2 W/cm^2^ (black), 0.41 W/cm^2^ (blue) and 0.6 W/cm^2^ (red). (**D**) Cycles of “on-off” laser at 808 nm (0.41 W/cm^2^) for TYMV−IONP colloid (1 mg_Fe_/mL).

**Figure 7 nanomaterials-13-02509-f007:**
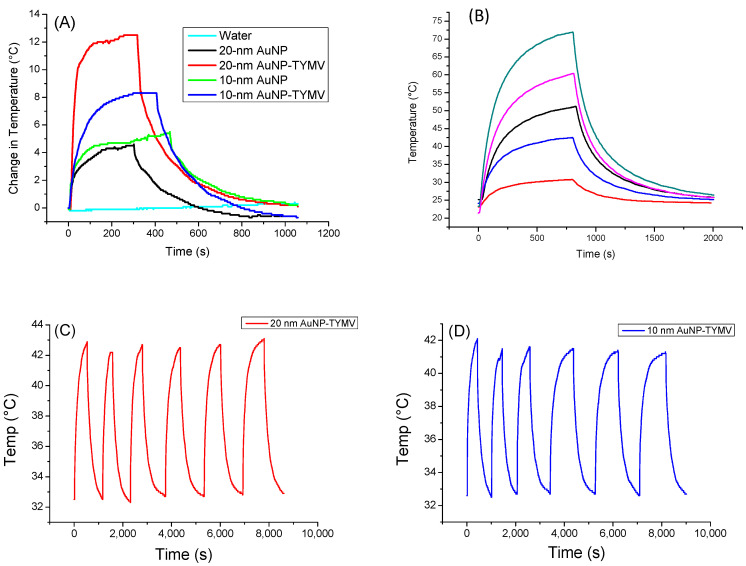
Photothermal experiments of AuNP and TYMV−AuNP aqueous colloids. (**A**) Temperature curve of AuNP and TYMV−AuNP colloids under the excitation of 808 nm with laser on–laser off (0.41 W/cm^2^). (**B**) Temperature curve of 20-nm AuNP−TYMV colloid under different power density laser at 808 nm: 0.165 W/cm^2^ (red), 0.536 W/cm^2^ (blue), 0.78 W/cm^2^ (black), 1.12 W/cm^2^ (pink), 1.5 W/cm^2^ (dark green). Cycles of “on-off” laser at 808 nm (0.41 W/cm^2^) for TYMV−AuNP made from (**C**) 20 nm- and (**D**) 10 nm-sized gold particles (0.02 mg_Au_/mL).

**Figure 8 nanomaterials-13-02509-f008:**
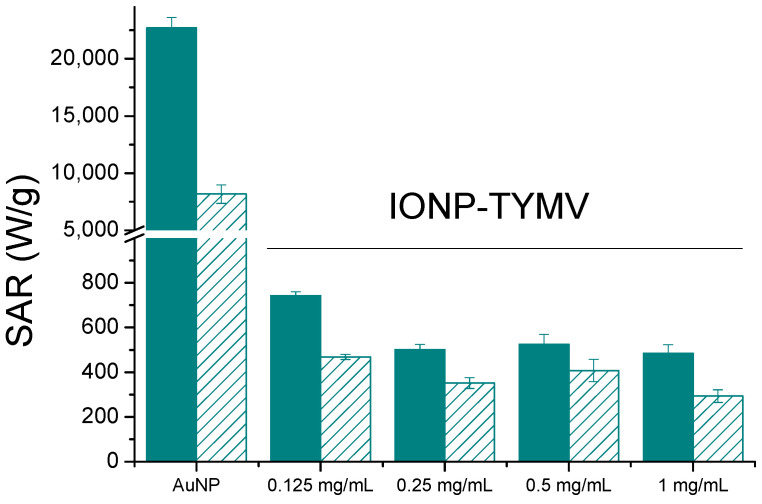
Heating capacities, SAR (W/g_Au_ or W/g_Fe_) for free, about 10 nm-sized Au and Fe_2_O_3_ particles (unfilled) or assembled onto TYMV (filled) at different concentrations (808 nm, 0.41 W/cm^2^).

## Data Availability

Data is contained within the article or Appendix A.

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
