# Peer review of "Gold and Iron Oxide Nanoparticle Assemblies on Turnip Yellow Mosaic Virus for In-Solution Photothermal Experiments"

_nanomaterials, 2023, doi:10.3390/nano13182509_

Round 1
Reviewer 1 Report
Nguyen et al. used Turnip yellow mosaic virus (TYMV) and grafted onto its outer surface gold nanoparticles (AuNP) or iron oxide nanoparticles (IONP). The resulting biohybrids have good heating capability revealing that these virus-based materials could be used as photothermal therapeutic agents. Overall, it is believed that the work has some novelty, and the data can support the conclusion. Before the paper is accepted, several concerns need to be addressed.
1. Why the photothermal response rate is low and generally takes several hundred minutes to reach its maximum? How to improve the response rate?
2. Which is more suitable for gold or iron oxide nanoparticles on Turnip yellow mosaic virus for photothermal application? How much difference is there? What is the reason?
There are no major issues with writing
Reviewer 2 Report
Photothermal therapy is a treatment method that uses materials with high photothermal conversion efficiency, injects them into tumor tissue, and converts light energy into heat energy under the irradiation of an external light source to kill cancer cells. As the first generation of photothermal therapy materials, metal nanoparticles have been in the upsurge of research. In this manuscript, Ha Anh Nguyen et al. used Turnip yellow mosaic virus (TYMV) and grafted onto its outer surface gold nanoparticles (AuNP) or iron oxide nanoparticles (IONP), and evidenced a net enhancement of the heating capability of these systems compared to their free NP counterparts. It provides views that these virus-based materials could be used as photothermal therapeutic agents. However, this manuscript has a lot of room for improvement in content and figures.
1. In “Introduction”, there is no causal relationship between the sentence “Thus, cell uptaken plasmonic nanoparticles, like gold nanoparticles (AuNPs), , excited by a laser light induce local hyperthermia, known as photothermia” and the previous sentence. Please pay attention to the correct use of conjunctions in the sentence. Also, there is an extra comma after “(AuNPs)”, please delete it.
2. Judging from the article description, the applied advantages of this study are not outstanding. The authors should add some space in the introduction to emphasize the innovation of this work and the problem it addresses.
3. There must be a reason for the author to choose TYMV as the carrier for nanosynthesis, but there is almost no introduction about the characteristics of TYMV. Please explain the characteristics and advantages of TYMV. accordingly.
4. In “Characterization of IONPs and AuNPs”, this study tested the potential of IONPs and AuNPs, but there are no relevant result graphs to prove it, please add.
5. In Figure 1A, part of the ordinate was blocked, please adjust to make the picture fully presented.
6. There is a formatting problem with the order icon in the upper left corner of all the figures in the manuscript. First, their fonts and sizes are not uniform. Secondly, the location of the icons is not neat, especially the icons in Figure 4 are so confusing that it affects reading. Please revise carefully.
7. There are also some problems with the layout of the figures. Try to use a consistent format for the same type of result graph to make the overall figure more beautiful.
8. In Figure 7B, the image in the upper right corner is unclear and redundant, please review and make corrections.
9. The order in which the figures appear does not correspond to the description in the manuscript. For example, Figure 7A is on line 357, while Figure 7B is on line 350. And Figure 7A appears twice in two consecutive sentences. Please re-adjust the logical order of the article description or the combination of figures.
10. The writing logic and knowledge depth of this manuscript still need to be strengthened. It is recommended to refer to the following literature.(DOI: 10.1002/advs.202302208; DOI:10.1016/j.cej.2022.137889)
Moderate editing of English language required.
Reviewer 3 Report
The authors have applied TYMV onto AuNP and IONP, respectively. The optical and structural properties have demonstrated the successful preparations of TYMV-AuNP and TYMV-IONP. The authors have also demonstrated the photothermal performance of TYMV-AuNP and TYMV-IONP. Overall, this work can inspire more material design ideas of virus-based materials for the application as photothermia therapeutic agents. Therefore, I would like to recommend this work to publish in Nanomaterials. Below are some comments for the authors.
1. For IONP and IONP@CA, please provide the DLS spectra as Figure 2A.
2. For Figure 4, the caption is hard to understand. Please revise the caption of Figure 4. Furthermore, the authors have provided TEM images with different sizes in Figure 4. Please organize and revise.
3. For Figure 5, the spectra reveal different unit in y-axis. Please correct.
4. The unit of power density in Figure 6C should be corrected with superscript for “2”.
5. For the introduction “As excellent light absorbers through their plasmon resonance and heat conductors...”, more references could be cited to broaden the introduction.
https://doi.org/10.3390/nano10061123
Round 2
Reviewer 2 Report
It can be accepted.
Minor editing of English language required.
Reviewer 3 Report
The authors have addressed all issues raised by the reviewers. Therefore, I would like to recommend this manuscript to publish as its current form in Nanomaterials.